# Phosphodiesterase-5 inhibitors use and risk for mortality and metastases among male patients with colorectal cancer

Wuqing Huang [1✉], Jan Sundquist[1,2,3], Kristina Sundquist[1,2,3] & Jianguang Ji [1✉]

Phosphodiesterase-5 (PDE5) inhibitors are suggested to have anti-tumor effects and to inhibit surgery-induced immunosuppression. We aimed to explore whether post-diagnostic use of PDE5 inhibitors was associated with a better prognosis among male patients with colorectal cancer (CRC) and the role of open surgery in the association. Here we show that post-diagnostic use of PDE5 inhibitors is associated with a decreased risk of CRC-specific mortality (adjusted HR = 0.82, 95% CI 0.67-0.99) as well as a decreased risk of metastasis (adjusted HR = 0.85, 95% CI 0.74-0.98). Specifically, post-operative use of PDE5 inhibitors has a strong anti-cancer effect. The reduced risk of metastasis is mainly due to distant metastasis but not regional lymphatic metastasis. PDE5 inhibitors have the potential to be an adjuvant drug for patients with CRC to improve prognosis, especially those who have undergone open surgery.

[1] Center for Primary Health Care Research, Lund University, Region Skåne, Lund, Sweden. [2] Department of Family Medicine and Community Health, Department of Population Health Science and Policy, Icahn School of Medicine at Mount Sinai, New York, NY, USA. [3] Department of Functional Pathology, Center for Community-based Healthcare Research and Education (CoHRE), School of Medicine, Shimane University, Matsue, Japan. ✉email: Wuqing.huang@med.lu.se; Jianguang.ji@med.lu.se

Cancer statistics in 2019 reported that colorectal cancer (CRC) is the third most common newly diagnosed cancer with more than 14,000 new cases and is the third leading cause of death due to cancer with more than 50,000 CRC-specific deaths in the United States[1]. The prognosis of CRC has improved during the past decades but with a trend of slowing down according to recent data[1,2]. Surgery remains the first choice of therapy for patients with CRC[3]. However, accumulated evidence indicate that surgery, especially open surgery which is much more invasive, might increase the risk of new metastases or promote the outbreak of pre-existing micro-metastases[3,4]. Several mechanisms support this alarming hypothesis[3,5–8]. Among these mechanisms, surgery-induced immune suppression is well recognized and considered as the critical process that leads to adverse outcomes among patients with cancer post surgery[8]. In recent years, myeloid-derived suppressor cells (MDSCs) are a highly heterogeneous population of immature myeloid cells that are becoming the high-value target in the field of cancer therapy[9–12]. Tumor-bearing mice model found that the level of MDSCs is increased significantly after cancer surgery, whereas cytotoxic T lymphocytes (CTLs), natural killer (NK) cells, and dendritic cells are decreased significantly[5,13,14].

Phosphodiesterase-5 (PDE5) inhibitors are a series of drugs used as the mainstay of treatment for erectile dysfunction, and also prescribed for patients who are diagnosed with pulmonary arterial hypertension[15]. Emerging evidence from in vivo and in vitro experiments suggest that PDE5 inhibitors might have an anticancer effect[15–27]. Sildenafil could inhibit colonic tumorigenesis via blocking the recruitment of MDSCs as shown by a recent study based on DSS-induced inflammation mice model[17]. In addition, an experiment conducted in vitro using human CRC cell observed that sildenafil could inhibit metastases by restoring the cytotoxicity of MDSCs-dependent inhibition of NK cell[22].

Based on the evidence mentioned above, we hypothesized that inhibition of PDE5 might be associated with a reduced risk of tumor progression and mortality among patients with CRC, and the effect might be stronger among patients who have received open surgery. However, population-based evidence is still lacking. We firstly access the TCGA Colon and Rectal Cancer (COAD-READ) cohort, and demonstrate a significant association between PDE5A gene expression and overall survival. Next, by combining several national databases in Sweden, we aim to explore the antitumor effects of PDE5 inhibitors in a population-based cohort and demonstrate that: (1) post-diagnostic use of PDE5 inhibitors is associated with a lower CRC-specific mortality; (2) post-diagnostic use of PDE5 inhibitors could reduce the incidence of metastasis among male patients who were diagnosed with CRC; and (3) the anticancer effect of PDE5 inhibitors is stronger among patients who received open surgery.

## Results

### Association of PDE5A expression with survival in patients with CRC.
Data from 430 patients with CRC from TCGA dataset showed that patients with a high expression of PDE5A had a significant poorer survival than patients with a low expression (Fig. 1, P value = 0.03972).

### Comparison of baseline characteristics between PDE5 inhibitors users and non-users.
By linking to the Swedish Cancer Registry and the Swedish Prescribed Drug Register, we identified a total of 12,465 male patients diagnosed with CRC at Stage I, II, or III during the study period. Among them, a total of 1136 patients had previously used PDE5 inhibitors after the diagnosis of CRC. Table 1 shows the baseline demographic and clinical factors among patients who used PDE5 inhibitors and those that

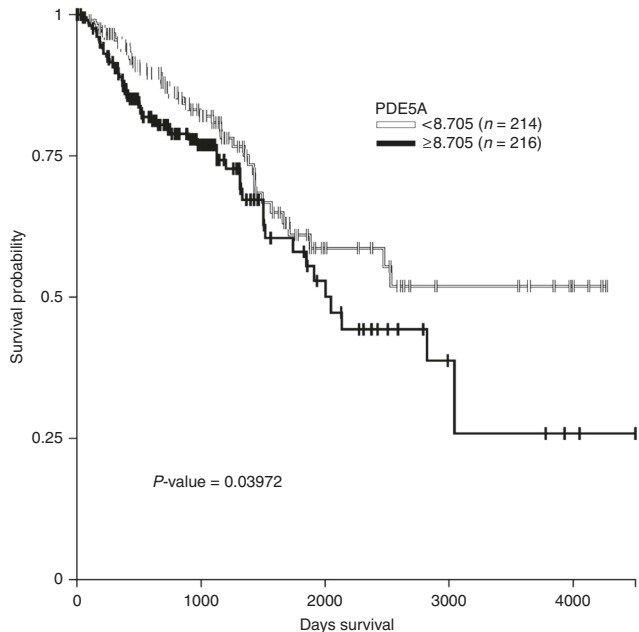

**Fig. 1 Overall survival curves stratified by expression of PDE5A gene.** Log-rank test was used to compare the difference of survival curves. Data of PDE5A gene expression in patients with colorectal cancer were collected from TCGA colon and rectal cancer, including 214 patients with low expression and 216 patients with high expression for survival curve plot, respectively.

did not. Compared with patients that did not use PDE5 inhibitors, post-diagnostic users were diagnosed at a younger age and at an earlier stage. Patients who used PDE5 inhibitors were more likely to get married and had a higher education and income level than patients who did not use PDE5 inhibitors. In addition, PDE5 inhibitor users were less likely to be dispensed with aspirin, and had a lower prevalence of comorbidity at baseline (Charlson Comorbidity Index (CCI) = 0).

### Association of PDE5 inhibitors use with mortality in male patients with CRC.
A significant difference of cause-specific survival curve was found between patients with and without PDE5 inhibitors (Fig. 2, P value < 0.001). As shown in Table 2, after a median follow-up of 4.25 years, 116 patients who used PDE5 inhibitors had died from CRC during the study period thus generating the mortality rate of 20.20 each 1000 person-year. By contrast, after a median follow-up of 4.25 years, the mortality rate was 37.87 each 1000 person-year among patients who did not use PDE5 inhibitors. After adjustment for potential confounders, post-diagnostic use of PDE5 inhibitors was associated with a decreased risk of death due to CRC (adjusted hazard ratios (HR) = 0.82, 95% confidence intervals (CI) = 0.67–0.99). The observed association was more pronounced in patients who underwent open surgery after diagnosis (adjusted HR = 0.72, 95% CI = 0.56–0.93), especially those with dispensation after surgery (adjusted HR = 0.69, 95% CI = 0.50–0.96). The association was not significant among patients who did not undergo open surgery (adjusted HR = 1.04, 95% CI = 0.78–1.40). Besides, the negative association was stronger among patients diagnosed with CRC at earlier stage than those with advanced stage (adjusted HR$_{\text{stage I/II vs. stage III}}$: 0.77 vs. 0.86). A dose–response analysis indicated a significant nonlinear relationship (P < 0.001) (Fig. 3).

### Association of PDE5 inhibitors use with metastasis in male patients with CRC.
As shown in Table 3, a total of 230 patients

| Table 1 Baseline characteristics of male patients with colorectal cancer. | | | | |
|---|---|---|---|---|
| Characteristics | Post-diagnostic users of PDE5 inhibitors (number, %) | | Without PDE5 inhibitors (number, %) | |
| Overall | 1136 | 100 | 11,329 | 100 |
| Age at diagnosis | | | | |
| <66 | 624 | 54.9 | 3416 | 30.1 |
| 66–75 | 397 | 35.0 | 3740 | 33.0 |
| >75 | 115 | 10.1 | 4173 | 36.8 |
| Year of diagnosis | | | | |
| 2005–2010 | 847 | 74.6 | 7178 | 63.4 |
| 2011–2014 | 289 | 25.4 | 4151 | 36.6 |
| Cancer stage at diagnosis | | | | |
| Stage I | 368 | 32.4 | 2886 | 24.5 |
| Stage II | 401 | 35.3 | 4451 | 39.3 |
| Stage III | 367 | 32.3 | 3992 | 35.2 |
| Birth country | | | | |
| Sweden | 991 | 87.2 | 9938 | 87.7 |
| Others | 145 | 12.8 | 1391 | 12.3 |
| Marital status | | | | |
| Married | 752 | 66.2 | 7257 | 64.1 |
| Unmarried | 121 | 10.7 | 1723 | 15.2 |
| Widom | 207 | 18.2 | 1541 | 13.6 |
| Divorce | 56 | 4.9 | 808 | 7.1 |
| Highest education | | | | |
| 1–9 years | 307 | 27.0 | 4557 | 40.2 |
| 10–11 years | 426 | 37.5 | 4170 | 36.8 |
| 12+ years | 403 | 35.5 | 2602 | 23.0 |
| Individual disposable income | | | | |
| Lowest | 161 | 14.2 | 2857 | 25.2 |
| Middle-low | 179 | 15.7 | 2904 | 25.6 |
| Middle-high | 299 | 26.3 | 2848 | 25.2 |
| Highest | 497 | 43.8 | 2720 | 24.0 |
| Prescription of other medicines | | | | |
| Aspirin | 262 | 23.1 | 3611 | 31.9 |
| Steroid | 367 | 32.3 | 3707 | 32.7 |
| Statins | 53 | 4.7 | 603 | 5.3 |
| Mental diseases | | | | |
| Depression | 19 | 1.7 | 176 | 1.6 |
| Anxiety | 8 | 0.7 | 101 | 0.9 |
| Charlson Comorbidity Index | | | | |
| 0 | 922 | 81.2 | 7978 | 70.4 |
| 1 | 174 | 15.3 | 2394 | 21.1 |
| 2 | 33 | 2.9 | 711 | 6.3 |
| >2 | 7 | 0.6 | 246 | 2.2 |

who had used PDE5 inhibitors suffered metastasis with an incidence rate of 42.20 each 1000 person-year, while the rate was 51.40 each 1000 person-year among those patients that did not use PDE5 inhibitors. Compared with patients who didn't use PDE5 inhibitors, post-diagnostic use of PDE5 inhibitors was significantly associated with a reduced risk of metastasis (adjusted HR = 0.85, 95% CI = 0.74–0.98). The decreased risk of metastasis was also more dominant among patients who underwent open surgery (adjusted HR = 0.74, 95% CI = 0.60–0.90) as compared with patients that did not undergo surgery (adjusted HR = 1.10, 95% CI = 0.90–1.35). We further investigated the association between post-diagnostic use of PDE5 inhibitors and locations of metastasis (Table 3). A significant inverse association was observed between PDE5 inhibitors use and the risk of distant metastasis with an adjusted HR of 0.83 (95% CI = 0.71–0.96). No significant association was found between PDE5 inhibitors use and regional lymphatic metastasis (adjusted HR = 1.12, 95% CI = 0.77–1.62).

**Sensitivity analyses**. Results of the sensitivity analyses are shown in Supplementary Table 1. In the first sensitivity analysis, after excluding patients who had ever previously used PDE5 inhibitors, we examined the association between post-diagnostic use of alprostadil and the risk of metastasis and CRC-specific mortality. No significant association was found between alprostadil use and risk of metastasis and CRC-specific mortality. In the second analysis, the results remained similar when compared with the matched comparisons (adjusted HR for CRC-specific mortality = 0.76, 95% CI = 0.62–0.93; adjusted HR for metastasis = 0.88, 95% CI = 0.76–1.02). The association was also stronger in patients with open surgery as shown in Supplementary Table 2 (adjusted $HR_{death}$ = 0.68; adjusted $HR_{metastasis}$ = 0.78). In the third analysis, after taking the competing risk into consideration, PDE5 inhibitors use in patients with open surgery was associated with a nonsignificant lower risk of mortality and a significant lower risk of metastasis (adjusted $HR_{death}$ = 0.81, 95% CI = 0.63–1.03; adjusted $HR_{metastasis}$ = 0.80, 95% CI = 0.66–0.98) (Supplementary Table 2). In the fourth analysis, we investigated the association between PDE5 inhibitors use and the prognosis among patients without comorbidity at diagnosis. The protective effect of PDE5 inhibitors was observed among patients with open surgery (adjusted $HR_{death}$ = 0.85; adjusted $HR_{metastasis}$ = 0.80) (Supplementary Table 2). In the fifth analysis, the association was even stronger after including patients diagnosed with CRC at stage IV or unknown stage (adjusted $HR_{death}$ = 0.70; adjusted $HR_{metastasis}$ = 0.80). After excluding patients with no more than 6 months of follow-up in the sixth analysis, the observed association with mortality or metastasis was comparable with the main results (adjusted $HR_{death}$ = 0.82; adjusted $HR_{metastasis}$ = 0.81). In seventh sensitivity analysis, patients with solely postoperative use of PDE5 inhibitors were at a lower risk of metastasis (adjusted $HR_{metastasis}$ = 0.52) when compared with patients without use of PDE5 inhibitors.

## Discussion
Higher expression of PDE5A gene is found to be significantly associated with a poorer survival among patients with CRC in COADREAD cohort, indicating a novel chemotherapeutic target for CRC treatment. Thus we explored the therapeutic potential of PDE5 inhibitors in a retrospective cohort of patients with CRC, and we find that post-diagnostic use of PDE5 inhibitors could significantly reduce the subsequent development of metastasis as well as the risk of death due to CRC. The negative association shows a nonlinear dose–response relationship. In particular, administration of PDE5 inhibitors after the open surgery shows the strongest protective effect, with a 39% decrease of the risk of CRC-specific mortality and a 31% decrease of the risk of metastasis, whereas patients who did not undergo open surgery did not show a significant association. Further analyses stratified by different types of metastasis indicate that post-diagnostic use of PDE5 inhibitors was significantly associated with a reduced risk of distant metastasis but not with regional lymphatic metastasis, which suggests that the preventive effect of PDE5 inhibitors on metastasis might be due to the inhibition of surgery-induced immunosuppression and recovering the function of CTLs and NK cells.

PDE5 inhibitors, including sildenafil, tadalafil, and vardenafil, are widely used around the world by patients to treat erectile dysfunction and have fewer side effects[15]. The antitumor effect of PDE5 inhibitors has been investigated in vivo or in vitro experiments, including colorectal, breast, and lung cancers[15–24]. Two recent clinical trials, conducted among patients with head and neck squamous cell carcinoma, reported that tadalafil can enhance systematic immune responsiveness as well as

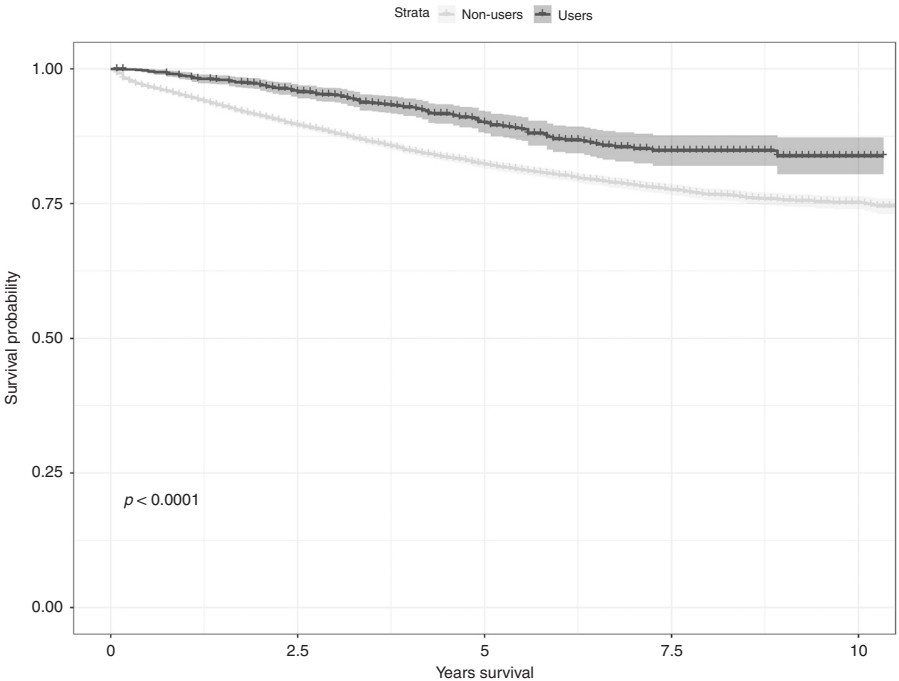

**Fig. 2 Cause-specific survival curves stratified by PDE5 inhibitors use.** Log-rank test was used to compare the difference of cause-specific survival curves. Shadow part indicates the 95% CI for the survival probability. 1136 patients using PDE5 inhibitors and 11,329 patients without using PDE5 inhibitors were included for survival curve plot, respectively.

tumor-specific immunity by reducing MDSCs, regulatory T cells and improving T-cell function[28,29].

Although it is still unclear about the potential mechanisms of the antitumor effect of PDE5 inhibitors, several hypotheses have been put forward, including restoring immunosuppression, increasing chemotherapeutic sensitization and permeability through blood–brain barrier, reversing hypoxia-induced resistance, and the induction of apoptosis[15]. A recent experiment using melanoma-bearing mice that underwent major surgery found that perioperative usage of PDE5 inhibitors reduced postoperative lung metastatic disease by restoring the function of NK cell[17]. The authors further performed an experiment in vitro and observed that MDSCs in the peripheral blood of CRC patients were significantly accumulated following cancer surgery. Co-culturing surgery-derived MDSCs with NK cells found that the functions of NK cells were impaired and the ability of tumor lysis was significantly reduced. Interestingly, MDSCs-dependent NK cell cytotoxicity was restored in the presence of PDE5 inhibitors[17]. Findings from this population-based study support the hypothesis that PDE5 inhibitors could improve the prognosis among patients with CRC through modulating surgery-induced immune suppression. Compared with CRC patients that did not use PDE5 inhibitors, the decreased risk of metastasis as well as CRC-specific mortality is most prominent among patients who were dispensed with PDE5 inhibitors after open surgery.

Further analyses targeting the locations of metastasis indicate that PDE5 inhibitors played a protective role in preventing distant metastasis but not regional lymphatic metastasis. This findings offer extra evidence for the hypothesis that the underlying anticancer mechanisms of PDE5 inhibitors may be due to recovering MDSCs-related immunosuppression. Previous studies indicated that MDSCs might play a critical role in distant metastasis in combination with the activities of circulating tumor cells (CTC)[11]. CTCs are cells that have shed into peripheral blood from a primary tumor and are carried around the whole body[30]. Growing evidence indicates that CTCs in the peripheral blood is

the prerequisite of distant metastasis, which is a multistage process[30]. Following the releasing from primary tumor, CTCs continue to survive in the blood, colonize at distant organs, and finally grow into a new tumor[30]. MDSCs were found to facilitate CTCs invasion by the secretion of matrix metalloproteinases and epithelial–mesenchymal transition, and promote CTCs immune evasion in circulatory system by impairing the function of NK cell and T cell[31]. Based on the data presented from this study, PDE5 inhibitors could be a novel postoperative adjuvant therapy for patients with CRC by restoring surgery-induced MDSCs-mediated immunosuppression.

There are some strengths and limitations of this population-based study. The major strength is the cohort study design with nationwide coverage to explore the anticancer effect of PDE5 inhibitors among patients diagnosed with CRC by linking several Swedish registers. Data retrieved from national registers with high quality and completeness of the follow-up allowed the study to exclude recall bias, mitigate selection bias, and avoid the causal reverse. In addition, the nationwide coverage makes it available to evaluate the dose–response relationship, which offers a strong evidence for the observed association. A few demographic and clinical factors, especially clinical stage at diagnosis, which is the most important factor that affects the prognosis of CRC, were included in our regression models to control their potential confounding effect. The confounding by indication was controlled by exploring the associations of alprostadil with metastasis and CRC mortality. Alprostadil is another drug used for the treatment of erectile dysfunction in Sweden, and our results suggest that confounding by indication might be minimal. An important limitation in the current study is that the study is limited to male patients because only a few female patients were prescribed with PDE5 inhibitors in Sweden. However, it might be worthwhile to explore whether the observed findings can be replicated in female patients with CRC as female patients might have a better tolerance of PDE5 inhibitors. In addition, some individual-level risk factors are not available in our study such as

**Table 2 Association of mortality due to colorectal cancer with PDE5 inhibitors.**

| Characteristics | No. of patients | Person-years | No. of deaths | IR, 1000 person-years | Crude HR | 95% CI | P value | Adjusted HR[a] | 95% CI | P value |
|---|---|---|---|---|---|---|---|---|---|---|
| **Post-diagnostic use of PDE5 inhibitors[b]** | | | | | | | | | | |
| No | 11,329 | 52,475 | 1987 | 37.87 | 1.00 | - | - | 1.00 | - | - |
| Yes | 1136 | 5743 | 116 | 20.20 | 0.62 | 0.52–0.75 | <0.001 | 0.82 | 0.67–0.99 | 0.038 |
| **Open surgery in colorectum** | | | | | | | | | | |
| Without open surgery | | | | | | | | | | |
| Without PDE5 inhibitors | 3606 | 14,280 | 891 | 62.40 | 1.00 | - | - | 1.00 | - | - |
| Post-diagnostic use | 299 | 1378 | 50 | 36.28 | 0.75 | 0.57–1.01 | 0.055 | 1.04 | 0.78–1.40 | 0.767 |
| Open surgery | | | | | | | | | | |
| Without PDE5 inhibitors | 7723 | 38,195 | 1096 | 28.69 | 1.00 | - | - | 1.00 | - | - |
| Post-diagnostic use | 837 | 4365 | 66 | 15.12 | 0.56 | 0.43–0.71 | <0.001 | 0.72 | 0.56–0.93 | 0.013 |
| Preoperative use[c] | 336 | 1783 | 27 | 15.14 | 0.58 | 0.39–0.84 | 0.005 | 0.76 | 0.52–1.12 | 0.166 |
| Postoperative use[d] | 501 | 2705 | 39 | 14.42 | 0.54 | 0.39–0.75 | <0.001 | 0.69 | 0.50–0.96 | 0.028 |
| **Stage at diagnosis** | | | | | | | | | | |
| Stage I or II | | | | | | | | | | |
| Without PDE5 inhibitors | 7337 | 35,498 | 853 | 24.03 | 1.00 | - | - | 1.00 | - | - |
| Post-diagnostic use | 769 | 4007 | 46 | 11.48 | 0.55 | 0.41–0.74 | <0.001 | 0.77 | 0.57–1.04 | 0.089 |
| Stage III | | | | | | | | | | |
| Without PDE5 inhibitors | 3992 | 16,977 | 1134 | 66.80 | 1.00 | - | - | 1.00 | - | - |
| Post-diagnostic use | 367 | 1736 | 70 | 40.33 | 0.71 | 0.56–0.91 | 0.006 | 0.86 | 0.67–1.10 | 0.219 |

Time-dependent Cox regression was used to calculate HRs and 95% CIs.
[a]Adjusted for age at diagnosis, year of diagnosis, cancer stage at diagnosis, birth country, marital status, highest education, individual disposable income, prescription of other medicines (including aspirin, steroid, and statin), mental diseases (including depression and anxiety), and Charlson Comorbidity Index.
[b]Post-diagnostic use of PDE5 inhibitors was defined as ≥2 prescriptions after diagnosis with colorectal cancer.
[c]Preoperative use of PDE5 inhibitors was defined as last use before open surgery in the colorectum.
[d]Postoperative use of PDE5 inhibitors was defined as last use after open surgery in the colorectum.

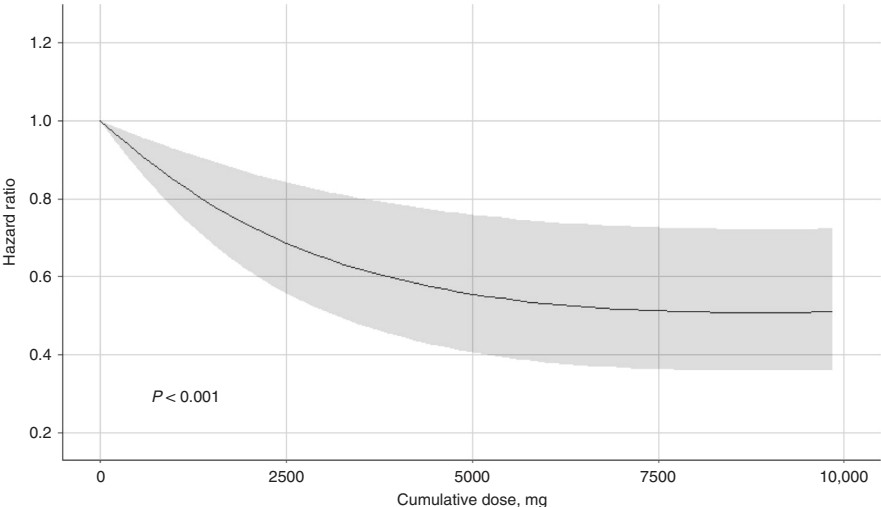

**Fig. 3 Dose–response association of PDE5 inhibitors with CRC-specific mortality.** Colorectal cancer CRC, Hazard ratio HR, Confidence interval CI. Cox regression model by using restricted cubic splines with three knots was used to assess dose–response relationship. HR was adjusted for age at diagnosis, year of diagnosis, cancer stage at diagnosis, birth country, marital status, highest education, income, prescription of other medicines (including aspirin, steroid, and statin), depression, and Charlson Comorbidity Index. Shadow part indicates the 95% CI for the HR.

| Characteristics | No. of patients | Person-years | No. of metastases | IR, 1000 person-years | Crude HR | 95% CI | P value | Adjusted HR[a] | 95% CI | P value |
|---|---|---|---|---|---|---|---|---|---|---|
| **Table 3 Association of metastasis with PDE5 inhibitors.** | | | | | | | | | | |
| **Post-diagnostic use of PDE5 inhibitors[b]** | | | | | | | | | | |
| No | 11,329 | 49,082 | 2523 | 51.40 | 1.00 | – | – | 1.00 | – | – |
| Yes | 1136 | 5450 | 230 | 42.20 | 0.84 | 0.73–0.96 | 0.014 | 0.85 | 0.74–0.98 | 0.026 |
| **Open surgery in the colorectum** | | | | | | | | | | |
| Without open surgery | | | | | | | | | | |
| Without PDE5 inhibitors | 3606 | 11,600 | 1427 | 123.02 | 1.00 | – | – | 1.00 | – | – |
| Post-diagnostic use | 299 | 1143 | 123 | 107.60 | 0.99 | 0.82–1.21 | 0.945 | 1.10 | 0.90–1.35 | 0.333 |
| Open surgery | | | | | | | | | | |
| Without PDE5 inhibitors | 7723 | 37482 | 1096 | 29.24 | 1.00 | – | – | 1.00 | – | – |
| Post-diagnostic use | 837 | 4307 | 107 | 24.84 | 0.73 | 0.59–0.89 | 0.002 | 0.74 | 0.60–0.90 | 0.003 |
| Preoperative use[c] | 336 | 1637 | 41 | 25.05 | 0.75 | 0.55–1.03 | 0.078 | 0.74 | 0.54–1.02 | 0.745 |
| Postoperative use[d] | 501 | 2670 | 66 | 24.72 | 0.71 | 0.55–0.91 | 0.007 | 0.73 | 0.57–0.95 | 0.017 |
| **Locations of metastasis** | | | | | | | | | | |
| Regional lymphatic metastasis | | | | | | | | | | |
| Without PDE5 inhibitors | 11,329 | 49,082 | 452 | 9.21 | 1.00 | – | – | 1.00 | – | – |
| Post-diagnostic use | 1136 | 5450 | 36 | 6.61 | 1.12 | 0.78–1.62 | 0.545 | 1.12 | 0.77–1.62 | 0.561 |
| Distant metastasis | | | | | | | | | | |
| Without PDE5 inhibitors | 11,329 | 49,082 | 2137 | 43.54 | 1.00 | – | – | 1.00 | – | – |
| Post-diagnostic use | 1136 | 5450 | 201 | 36.88 | 0.81 | 0.70–0.94 | 0.005 | 0.83 | 0.71–0.96 | 0.014 |

Time-dependent Cox regression was used to calculate HRs and 95% CIs.
[a]Adjusted for age at diagnosis, year of diagnosis, cancer stage at diagnosis, birth country, marital status, highest education, individual disposable income, prescription of other medicines (including aspirin, steroid, and statin), mental diseases (including depression and anxiety), and Charlson Comorbidity Index.
[b]Post-diagnostic use of PDE5 inhibitors was defined as ≥2 prescriptions after diagnosis with colorectal cancer.
[c]Preoperative use of PDE5 inhibitors was defined as last use before open surgery in the colorectum.
[d]Postoperative use of PDE5 inhibitors was defined as last use after open surgery in the colorectum.

dietary factors and medical treatments of CRC, which may have some confounding effect. However, Sweden is well-known for its universal healthcare system; discrepancy in medical treatment is relatively uncommon[32]. Moreover, we have adjusted for income and education level in the multiple regression model, which might help to control the confounding by dietary factors. It is uncertain whether patients who dispersed the medicines from the pharmacy would comply with the treatment regimen. To take into consideration the possibility of noncompliance, only patients who were dispensed with PDE5 inhibitors more than one time are defined as post-diagnostic users. Uncertainty of medical compliance may induce exposure misclassifications, which, however, might lead to underestimating the observed inverse association in this study.

In conclusion, our study find a significant anticancer effect of PDE5 inhibitors as shown by a reduced risk of metastasis and a lower risk of CRC-specific mortality. It is noteworthy that post-operative use of PDE5 inhibitors have a stronger protective effect, which supports the hypothesis that the PDE5 inhibition might perform antitumor effect via regulating immunosuppression, especially surgery-induced immunosuppression. PDE5 inhibitors might be a potential adjuvant drug for patients with CRC to improve prognosis, especially, those who have undergone open surgery.

## Methods

**Study population.** The Ethics Committee at Lund University approved (February 6, 2013) this nationwide cohort study (Dnr 2012/795). Through advertisements in the major newspapers people could choose to opt out before the project databases were constructed. Written informed consent is not needed in Sweden for the register-based study. The project database is located at Center for Primary Health Care in Malmö, Sweden.

We identified all patients who were diagnosed with CRC as the first primary cancer between January 2005 and March 2014 from the Swedish Cancer Registry by using the 10th International Classification of Disease code (C18, C19, and C20)[33,34]. Only male patients were included in the present study as only a few female patients used PDE5 inhibitors in Sweden during the study period. Given that metastasis was the primary outcome in the study, we further excluded patients who were diagnosed with CRC at stage IV as cancer cells would have already spread to distant organs[35]. The TNM staging system, which includes the size of tumor (T), nodal status (N), and the presence of metastatic disease (M), was used to define the stage at diagnosis of CRC as stage I (T1 or T2 N0 M0), stage II (T3 or T4 N0 M0), stage III (any T N1 or N2 M0), and stage IV (any T or N M1)[2]. By linking the Swedish Cancer Registry to the National Patient Register (NPR), we were able to identify patients who had ever previously undergone open surgery in the colorectum by using the NOMESCO Classification of Surgical Procedures.

The following patients were excluded from this study if: (1) patients who received open surgery in the colorectum before the diagnosis of CRC; (2) patients who received open surgery in the colorectum after metastasis; (3) patients who were followed no more than 1 month; (4) patients who only used PDE5 inhibitors before the diagnosis of CRC or patients with only one dispensation of PDE5 inhibitors. We present the flowchart of patients included in this study in Supplementary Fig. 1.

**Assessment of PDE5 inhibitors use.** Information on use of PDE5 inhibitors among patients with CRC was retrieved from the Swedish Prescribed Drug Register. Anatomical Therapeutic Chemical (ATC) classification system was used to identify individuals who had been dispensed with PDE5 inhibitors, including sildenafil, tadalafil, and vardenafil by using ATC codes G04BE03, G04BE08, and G04BE09, respectively. Taking into consideration the possibility of non-adherence, we defined post-diagnostic use of PDE5 inhibitors as patients who had at least two dispensations after the diagnosis of CRC. We further categorized post-diagnostic PDE5 inhibitors use into preoperative use and postoperative use among patients who underwent open surgery. Preoperative use was defined as the date of last dispensation of PDE5 inhibitors, which was recorded before the open surgery; postoperative use was defined as the date of last dispensation of PDE5 inhibitors recorded after the open surgery, thus patients whose first dispensation happened before open surgery were also included. We aimed to explore whether post-operative use of PDE5 inhibitors might protect against mortality, thus all the patients who have ever administrated PDE5 inhibitors after the operation were included. We also explored the association among patients who used PDE5 inhibitors only after the operation (first dispensation of PDE5 inhibitors after open surgery), as shown in Supplementary Table 1. The cumulative dose of PDE5 inhibitors was calculated as the sum of the defined daily dose for all the dispensations during the follow-up period.

**Assessment of outcomes.** The primary outcome was death due to CRC as the primary cause of death. Data concerning the cause of death were collected from the Cause of Death Register with ICD-10 codes of C18, C19, and C20.

The secondary outcome was metastasis. We identified patients who were diagnosed with metastasis from the NPR with ICD-10 codes C77, C78, and C79. We further stratified the metastasis as regional lymphatic metastasis (C77.2 and C77.5) and distant metastasis.

**Assessment of covariates.** Information about covariates was listed in Table 1. Baseline demographic characteristics were collected through retrieving data from Statistics Sweden's Total Population Register and Population Housing Census. Marital status and individual disposable income were recorded at 2005. According to the quartiles among patients without PDE5 inhibitors, individual disposable income was grouped as lowest, middle-low, middle-high, and highest. Country of birth was modeled as born in Sweden or not. Highest education level was modeled as 1–9 years, 10–11 years, and 12+ years. Clinical data related to CRC were obtained from the Swedish Cancer Registry. We further retrieved other clinical information that was associated with metastasis and mortality from the Swedish Prescribed Drug Register and the NPR, including prescription of aspirin, steroid and statins, and diagnosis of mental diseases (depression and anxiety) at baseline, which were suggested to be more common among patients with erectile dysfunction as compared with the general population. As having a comorbid condition is an important factor affecting the prognosis in patients with CRC, we calculated the CCI based on a total of 22 conditions at baseline, including heart diseases, AIDS, and so on (CRC was not included in the calculation)[36]. The index was modeled as 0, 1, 2, and >2 indicating the most healthy to the worst condition.

All linkages were performed by the individual national identification number, which is assigned to all residents staying in Sweden longer than 3 months (residence permit). To preserve confidentiality, this ID number was replaced by a serial number.

**Assessment of PDE5A expression.** The association between the overall survival and expression of PDE5A was analyzed by accessing COADREAD cohort (http://xena.ucsc.edu/)[37]. Basic clinical characteristics were retrieved from TCGA colon adenocarcinoma and rectum adenocarcinoma datasets. The gene expression was measured using the Illumina HiSeq 2000 RNA Sequencing platform by the University of North Carolina TCGA genome characterization center. Gene-level transcription estimates were shown as $\log2(x + 1)$ transformed RSEM normalized count. The samples were divided into two groups based on the median value 8.705. The follow-up started at the date of diagnosis and ended at date of death or the last time any individual was known to be alive. Log-rank test was used to compare the difference of Kaplan–Meier curves. The statistics from the Xena Browser reports were done using R's survival package of survdiff.

**Statistical analysis.** Considering the impact from immortal time bias, time-dependent Cox regression was used to calculate HRs and 95% CIs of CRC-specific mortality and metastasis associated with post-diagnostic use of PDE5 inhibitors. The follow-up started at the date of diagnosis with CRC (baseline), and ended at the time of occurrence of outcomes or at the end of the follow-up period (December 2015), whichever came first. Post-diagnostic use of PDE5 inhibitors was modeled as a time-dependent variable, thus allowing patients who moved from a follow-up period of non-exposure (from diagnosis of CRC to the second administration of PDE5 inhibitors) to a period of exposure (from the second administration of PDE5 inhibitors and thereafter for the remainder of follow-up). To account for the remaining imbalance, multiple regression analyses were conducted including covariates as follows: age at diagnosis, year of diagnosis, cancer stage at diagnosis, birth country, highest education, individual disposable income, marital status, prescription of other medicines (including aspirin, steroid and statin), mental diseases (including depression and anxiety), and CCI. Log-rank test was used to compare the difference of cause-specific survival curve between exposure and non-exposure, in which patients who used PDE5 inhibitors after CRC diagnosis were classified as unexposed person-time from diagnosis to the second dispensation and the subsequent person-time as exposed. Dose–response relationship was assessed in Cox regression model by using restricted cubic splines with three knots, which is allowed to intuitively represent a nonlinear relationship.

To explore whether the antitumor effect of PDE5 inhibitors was modified by open surgery, we did stratification analysis according to the history of open surgery in the colorectum after CRC diagnosis. Among patients with a history of open surgery, we further stratified patients into preoperative and postoperative use. Moreover, we stratified patients diagnosed at earlier stage (Stage I or II) and advanced stage (Stage III) to explore the association with CRC-specific mortality, as well as stratified the metastasis into regional lymphatic metastasis and distant metastasis.

Sensitivity analyses were further performed to explore the impacts due to potential bias. To assess the confounding by indication, in the first sensitivity analysis, we examined the prognosis in CRC patients who had used alprostadil

instead of PDE5 inhibitors for the treatment of erectile dysfunction as alprostadil was another common drug prescribed for patients with erectile dysfunction in Sweden. In the second sensitivity analysis, as time-dependent Cox regression could not totally control the confounding effect by immortal time bias, we adopted a matched cohort design as shown in Supplementary Fig. 2. Up to five patients who were not dispensed with PDE5 inhibitors and still alive on the date of second dispensation of PDE5 inhibitors of the corresponding patients (index date) were randomly matched with each patient with PDE5 inhibitors conditional on the same year of birth and year of CRC diagnosis. Follow-up started at the date of second dispensation of PDE5 inhibitors in users or the index date in the matched comparisons, and ended at the date of occurrence of outcomes or the end of the follow-up period, whichever came first. Taking into account the potential effect of competing risk, we further conducted Cox regression using competing risk model in the third sensitivity analysis, in which death due to other causes was served as a competing event. In the fourth sensitivity analysis, we investigated the association between PDE5 inhibitors and the prognosis among CRC patients without comorbidity (CCI = 0) to minimize the "healthy worker" effect. In the fifth one, male patients diagnosed with CRC at stage IV or unknown stage were included to explore the risk of post-diagnostic use of PDE5 inhibitors on the prognosis of CRC. In the sixth analysis, we examined the association after excluding patients with no more than six months of follow-up. In the seventh analysis, we further explore the protective effect of solely postoperative use (first dispensation of PDE5 inhibitors after open surgery) by comparing with patients with open surgery and without use of PDE5 inhibitors.

All analyses were performed using SAS version 9.4 or R 3.6.0. Statistical significance was set at $P$ value ≤ 0.05 (two-tailed) for all tests.

**Reporting summary.** Further information on research design is available in the Nature Research Reporting Summary linked to this article.

## Data availability
The data based on Swedish register are not publicly available due to Swedish law and protecting patients privacy, and the combined set of data used for the analysis presented in this study can only be made available from the appropriate Swedish authorities (the Swedish National Board of Health and Welfare (https://www.socialstyrelsen.se/en) and Statistics Sweden (https://www.scb.se/en), for researchers who meet the criteria for access to confidential. TCGA Colon and Rectal Cancer is combined from TCGA colon adenocarcinoma and rectum adenocarcinoma datasets, which can be accessed on UCSC Xena platform (https://tcga.xenahubs.net).

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

## Acknowledgements

The authors wish to thank the CPF's science editor Patrick Reilly for his valuable comments on the text. This work was supported by grants awarded to J.J. by the Swedish Research Council (2016-02373) and Cancerfonden (2017 CAN2017/340) and The Crafoord Foundation, to K.S. and to J.S. by the Swedish Research Council (2018-02400 and 2016-01176, respectively), to J.S., K.S., and J.J. by ALF funding from Region Skåne, and to W.H. by China Scholarship Council (Grant No. 201806380121). Open access funding provided by Lund University.

## Author contributions

W.H., J.J., K.S., and J.S. were responsible for the study concept and design. J.S., K.S., and J.J. contributed to obtain funding. K.S. and J.S. contributed to acquire the data. W.H. contributed to do the statistical analysis and drafted the manuscript, and all authors contributed to revise it for important intellectual content.

## Competing interests

The authors declare no competing interests.
