## [Peer Review File · Nature Communications]

Reviewers' comments:

Reviewer #1 (Remarks to the Author):

Huang et al. use Swedish register data to show that PDE5 inhibitors (otherwise known as Viagra) reduce the risk of colorectal cancer specific mortality and metastases in men diagnosed with colorectal cancer. This is a question that had not yet been addressed, and Swedish registers present a unique opportunity to provide an answer. I, however, have several concerns regarding the methodology and analysis:

1. The first results reported are that of the association between PDE5A expression and survival in patients with colorectal cancer. However, there is no extensive description of the methods used for this analysis. All that is mentioned in the methods is that the COADREAD cohort was used. The results then state that this only includes data from 430 patients. Who are these patients? Were they all also included in the main analysis? You show a Kaplan Meier plot, but don't explain when follow up started. Was it directly after colorectal cancer diagnosis? To understand these results, the reader needs further description of the methods.
2. It is not clear when follow up starts for individuals diagnosed with colorectal cancer included in the study. I assume that they are under follow up from the point of colorectal cancer diagnosis given the information provided when explaining exposure classification on page 13, lines 289-291. However, this should be made clearer.
3. Why was the National quality register for colon and rectal cancer not used? This would have enabled further detailed information on severity of colorectal cancer and cancer treatments used.
4. The authors state that patients were required to have at least two PDE5 inhibitor prescriptions after colorectal cancer diagnosis before exposure. Did exposure to PDE5 inhibitors then start at the date of second prescription, or was it predated to first prescription once they received their second prescription? This is not clear.
5. Was there any restriction on the time allowed between the two prescriptions? For example, could a patient have one PDE5 inhibitor prescription directly after colorectal cancer diagnosis, then another 18 months later and still be defined as being exposed?

6. Was date of dispensing or date of prescription used? The Swedish prescribed drug register only contains prescription and dispensing dates for those drugs that were actually dispensed. So, if prescription date was used, then there are likely some missing prescriptions i.e. people that were prescribed but did not dispense their PDE5 inhibitors. This should be discussed.

7. When defining exposure by pre- and post-operative use, only those patients whose last PDE5 inhibitor prescription was before the surgery were defined as being exposed prior to surgery. However, patients that were prescribed both before and after surgery were defined as post-operative users. Please can you explain the motivation behind this? Because the post-operative group includes a mix of individuals who were both pre- and post-operative users, and those who were solely post-operative users.

8. Why is does follow up for death end at March 2017, but follow up for metastases end at December 2015?

9. What happened to men that died of other causes during follow up? These deaths by other causes (such as CVD) could be competing events. Especially in the cases where PDE5 inhibitors were prescribed for hypertension.

10. The first line of the statistical analysis states that time-dependent Cox regression was used to reduce immortal time bias. This is not possible. The only way to stop immortal time bias is through carefully ensuring it does not exist in the study design. Time-updating exposure will not immediately stop immortal time bias.

11. As touched on above in point 7, authors time-updated the exposure. However, it is not clear how confounders were defined. Were they only adjusted for at baseline (time of colorectal cancer diagnosis)? Please make this clear. If time-updating exposure, it is also important to think if there is any treatment-confounder feedback i.e. do time-varying confounders change the likelihood of changing exposure? If so, then merely time-updating confounder is not enough. More advanced methods such as inverse probability weighting or the g-formula need to be used.

12. On page 13, line 306, authors state that sensitivity analyses were performed to control potential biases. It is not possible to control for biases through sensitivity analyses. You can only explore if potential biases will impact our results.

13. There is no need to include p-values in Table 1.

14. What was the average follow-up in those exposed and not exposed to PDE5 inhibitors? Hazard ratios give the average difference between the hazards of the two groups over all of follow up. If the follow-up differed between the two groups, then the difference in outcome could be a consequence of this difference. Providing simple Kaplan-Meier plots would also help to explore this.

Anthony Matthews, Karolinska Institutet.

Reviewer #2 (Remarks to the Author):

The authors examined the associations between post-diagnostic use of phosphodiesterase-5 (PDE5) inhibitors and CRC-specific mortality as well as metastasis among CRC patients from the Swedish Cancer Registry. The authors reported that post-diagnostic use of PDE5 inhibitors was associated with a decreased risk of CRC-specific mortality and metastasis. Some data presentations and analyses need to be shaped up substantially as suggested below.

1. Since the CRC patients and drug information are from two big pools, the Swedish Cancer Registry and the Swedish Prescribed Drug Register, and also the authors applied several inclusion and exclusion criteria to select the patients for final analysis (also sensitivity analyses), a clean flow-chart describing the process of the patients' selection (including the numbers occurred in each inclusion and exclusion criteria) would be helpful to the readers to better understand the study design.

2. For the evaluation of the use of PDE5 inhibitors, the authors included three inhibitors (sildenafil, tadalafil, and vardenafil). Actually, there are many other types of PDE5 inhibitors (e.g., Thiosildenafil, rac Xanthoanthrafil... etc.). Can the authors validate the reason that why they only did check three types (except the reason that these are commonly used drugs in Sweden). Some appropriate references might be helpful.

3. In addition to the frequency of drug use, the dose and duration of drug use are also important indices in terms of evaluation for the relationship between drug use and disease outcomes. However the duration of PDE5 inhibitors use has not been discussed (due to lack of information?). Also, the analysis and description regarding dose of drug use are not clearly described.

4. Considering that clinical stage at diagnosis is the most important factor that affects the CRC prognosis, drug use may have an inconsistent effect on the prognosis and metastasis for each stage of cancer. Did the authors get a chance to look at the associations stratified by the stage? Also, the authors excluded patients with CRC at stage IV for metastasis-related analysis; does the drug use affect the mortality for the patients with CRC at stage IV?

5. In the metastasis-related analyses, a sensitivity analysis after excluding the patients who died during follow-up would be worth to try in order to further examine the independent association with drug use.

6. English proofreading would be helpful to improve the quality of the manuscript.

Reviewers' comments:

Reviewer #1:

Huang et al. use Swedish register data to show that PDE5 inhibitors (otherwise known as Viagra) reduce the risk of colorectal cancer specific mortality and metastases in men diagnosed with colorectal cancer. This is a question that had not yet been addressed, and Swedish registers present a unique opportunity to provide an answer. I, however, have several concerns regarding the methodology and analysis:

1. The first results reported are that of the association between PDE5A expression and survival in patients with colorectal cancer. However, there is no extensive description of the methods used for this analysis. All that is mentioned in the methods is that the COADREAD cohort was used. The results then state that this only includes data from 430 patients. Who are these patients? Were they all also included in the main analysis? You show a Kaplan Meier plot, but don't

explain when follow up started. Was it directly after colorectal cancer diagnosis? To understand these results, the reader needs further description of the methods.

>>> We apologized for the unclear description. We have added detail information related to the study cohort, as well as the statistical analyses in the method section #Page 14. Patients in this analysis were retrieved from public data resources TCGA, who were not included in the main analyses using Swedish registers.

2. It is not clear when follow up starts for individuals diagnosed with colorectal cancer included in the study. I assume that they are under follow up from the point of colorectal cancer diagnosis given the information provided when explaining exposure classification on page 13, lines 289-291. However, this should be made clearer.

>>> Yes, the follow up started at the date of diagnosis with CRC. We added the description of follow up in the method section #Page 15 and display examples in Supplementary Figure 2 to make it clearer.

3. Why was the National quality register for colon and rectal cancer not used? This would have enabled further detailed information on severity of colorectal cancer and cancer treatments used.

>>> Thank the reviewer for pointing out this. We agree that the National quality register for colon and rectal cancer will add more information about severity of colorectal cancer and cancer treatments. Unfortunately, we did not get the permission to use it even we have tried to contact the owner of this register before.

4. The authors state that patients were required to have at least two PDE5 inhibitor prescriptions after colorectal cancer diagnosis before exposure. Did exposure to PDE5 inhibitors then start at the date of second prescription, or was it predated to first prescription once they received their second prescription? This is not clear.

>>> We apologized for being unclear of the definition. To overcome non-adherence of PDE5 inhibitors use, we identified patients with two or more dispensations of PDE5 inhibitors after diagnosis defined as users. Exposure to PDE5 inhibitors was defined as the date of first dispensation.

5. Was there any restriction on the time allowed between the two prescriptions? For example, could a patient have one PDE5 inhibitor prescription directly after colorectal cancer diagnosis, then another 18 months later and still be defined as being exposed?

>>> There is no restriction on the time interval between two prescriptions. The median value of time interval between first and second prescription in this study was 4 months.

6. Was date of dispensing or date of prescription used? The Swedish prescribed drug register only contains prescription and dispensing dates for those drugs that were actually dispensed. So, if prescription date was used, then there are likely some missing prescriptions i.e. people that were prescribed but did not dispense their PDE5 inhibitors. This should be discussed.

>>> Thank the reviewer pointing out this important question. Date of dispensing was used in the analysis, and we revised the description in the manuscript.

7. When defining exposure by pre- and post-operative use, only those patients whose last PDE5 inhibitor prescription was before the surgery were defined as being exposed prior to surgery. However, patients that were prescribed both before and after surgery were defined as post-operative users. Please can you explain the motivation behind this? Because the post-operative group includes a mix of individuals who were both pre- and post-operative users, and those who were solely post-operative users.

>>> We apologized for the unclear description about the definition of post-operative users, we revised it as shown in #Page 13. Post-operative use was defined as the date of last dispensation of PDE5 inhibitors recorded after the open surgery, thus patients whose first dispensation

happened before open surgery might be also included in this group. As we aimed to explore whether post-operative use of PDE5 inhibitors might protect against mortality, thus all the patients who have ever administrated PDE5 inhibitors after the operation were included. We also explored the association among patients who used PDE5 inhibitors only after the operation, as shown in sensitivity analysis 7.

8. Why is does follow up for death end at March 2017, but follow up for metastases end at December 2015?

>>> In our datasets, the information about overall death was updated to March 2017 from the Swedish Death Register. However, we have just noticed that the cause of death was updated only to December 2015. In addition, for the identification of metastasis from the Swedish Patient Register, it was updated to December 2015. To keep it consistent, we thus revised the follow up for CRC-specific death as December 2015 as well. Analyses have been redone accordingly.

9. What happened to men that died of other causes during follow up? These deaths by other causes (such as CVD) could be competing events. Especially in the cases where PDE5 inhibitors were prescribed for hypertension.

>>> We added a sensitivity analysis using competing risk model, and the result was shown in Sensitivity analysis 4 in Table 4.

10. The first line of the statistical analysis states that time-dependent Cox regression was used to reduce immortal time bias. This is not possible. The only way to stop immortal time bias is through carefully ensuring it does not exist in the study design. Time-updating exposure will not immediately stop immortal time bias.

>>> We agree with the reviewer that time-dependent Cox regression can not immediately stop immortal time bias. As Suissa et al mentioned, a Cox proportional hazards model with a time-

dependent definition for the drug exposure is a solution to reduce the impact from immortal time bias [1]. To further limit the contribution by immortal time bias, we additionally used matched cohort design. Up to five patients who were not dispensed with PDE5 inhibitors and still alive on the date of first dispensation of PDE5 inhibitors of the corresponding patients (index date) were randomly matched with each patient with PDE5 inhibitors conditional on the same year of birth and year of CRC diagnosis. Follow-up started at the date of first dispensation of PDE5 inhibitors in users or the index date in the matched comparisons, and ended at the date of occurrence of outcomes or the end of the follow-up period, whichever came first. The results were shown in Sensitivity analysis 3 in Table 4.

11. As touched on above in point 7, authors time-updated the exposure. However, it is not clear how confounders were defined. Were they only adjusted for at baseline (time of colorectal cancer diagnosis)? Please make this clear. If time-updating exposure, it is also important to think if there is any treatment-confounder feedback i.e. do time-varying confounders change the likelihood of changing exposure? If so, then merely time-updating confounder is not enough. More advanced methods such as inverse probability weighting or the g-formula need to be used.

>>> Thank the reviewer for this suggestion. In our original definition, we defined confounding factors as status before the end of follow up. We agree that it could be better to define confounders as status at baseline (time of colorectal cancer diagnosis). We re-defined these confounders in the revised version. All results related to multivariate model have been re-analyzed and the description has been revised in method section. Regarding for the potential treatment-confounder feedback caused by the time-updating exposure, we added a sensitivity analysis (Sensitivity analysis 3) in matched cohort design, where the treatment-confounder feedback was not supposed to exist.

12. On page 13, line 306, authors state that sensitivity analyses were performed to control potential biases. It is not possible to control for biases through sensitivity analyses. You can only explore if potential biases will impact our results.

>>> We totally agree with the reviewer and revised the sentence accordingly.

13. There is no need to include p-values in Table 1.

>>> It has been deleted.

14. What was the average follow-up in those exposed and not exposed to PDE5 inhibitors? Hazard ratios give the average difference between the hazards of the two groups over all of follow up. If the follow-up differed between the two groups, then the difference in outcome could be a consequence of this difference. Providing simple Kaplan-Meier plots would also help to explore this.

>>> The median follow up in users and non-users have been described in the results #Page 5, and Kaplan-Meier plots is present in Figure 2.

Reviewer #2:

The authors examined the associations between post-diagnostic use of phosphodiesterase-5 (PDE5) inhibitors and CRC-specific mortality as well as metastasis among CRC patients from the Swedish Cancer Registry. The authors reported that post-diagnostic use of PDE5 inhibitors was associated with a decreased risk of CRC-specific mortality and metastasis. Some data presentations and analyses need to be shaped up substantially as suggested below.

1. Since the CRC patients and drug information are from two big pools, the Swedish Cancer Registry and the Swedish Prescribed Drug Register, and also the authors applied several inclusion and exclusion criteria to select the patients for final analysis (also sensitivity analyses), a clean flow-chart describing the process of the patients' selection (including the numbers

occurred in each inclusion and exclusion criteria) would be helpful to the readers to better understand the study design.

>>> Thank you for the suggestion. We added a flowchart as shown in Supplementary Figure 1.

2. For the evaluation of the use of PDE5 inhibitors, the authors included three inhibitors (sildenafil, tadalafil, and vardenafil). Actually, there are many other types of PDE5 inhibitors (e.g., Thiosildenafil, rac Xanthoanthrafil... etc.). Can the authors validate the reason that why they only did check three types (except the reason that these are commonly used drugs in Sweden). Some appropriate references might be helpful.

>>> We agree with the reviewer that there are many other types of PDE5 inhibitors. However, only sildenafil, tadalafil, and vardenafil were used in Sweden and can be retrieved from the Swedish Prescribed Drug Register.

3. In addition to the frequency of drug use, the dose and duration of drug use are also important indices in terms of evaluation for the relationship between drug use and disease outcomes. However the duration of PDE5 inhibitors use has not been discussed (due to lack of information?). Also, the analysis and description regarding dose of drug use are not clearly described.

>>> We agree with the reviewer that the dose and duration of drug use are also important indices. We added the description about cumulative dose of drug use in method section #Page 13. Regarding for the duration of drug use, unlike some drugs, such as anti-hypertension which should be used continuously, the use of PDE5 inhibitors was randomly (which is usually taken only when needed, 30 minutes to 1 hour before sexual activity), so we could not determine the duration based on the Swedish Prescription Drug Register.

4. Considering that clinical stage at diagnosis is the most important factor that affects the CRC prognosis, drug use may have an inconsistent effect on the prognosis and metastasis for each

stage of cancer. Did the authors get a chance to look at the associations stratified by the stage? Also, the authors excluded patients with CRC at stage IV for metastasis-related analysis; does the drug use affect the mortality for the patients with CRC at stage IV?

>>>We agree that clinical stage at diagnosis is the most important factor, thus a stratified analysis by earlier or advance stage was added to explore the difference of associations between PDE5 inhibitors use and CRC-specific death. As patients with stage IV have already experienced metastasis, we thus excluded patients diagnosed at stage IV in the main analysis. However, we added a sensitivity analysis (sensitivity analysis 5) in Table 4 in which patients with stage IV and unknown stage were included. The protective effect of PDE5 inhibitors remained significant.

5. In the metastasis-related analyses, a sensitivity analysis after excluding the patients who died during follow-up would be worth to try in order to further examine the independent association with drug use.

>>> Death has the potential to compete the risk of metastasis, we thus add a sensitivity analysis (sensitivity analysis 4) using competing risk model. The results were shown in Table 4.

6. English proofreading would be helpful to improve the quality of the manuscript.

>>> Our scientific editor has gone through the paper.

References

1. Suissa S. Immortal time bias in pharmaco-epidemiology. *Am J Epidemiol* 167, 492-499 (2008).

Reviewers' comments:

Reviewer #1 (Remarks to the Author):

The authors have clearly put a lot of effort into revising the manuscript following my original comments. The revisions have considerably changed some of the original methodologies. For example, confounders have now been defined at baseline, rather than their status before the end of follow up. However, I still have some considerable worries about their methodological choices.

1) Following my original comment 4, the authors state: "To overcome non-adherence of PDE5 inhibitors use, we identified patients with two or more dispensations of PDE5 inhibitors after diagnosis defined as users. Exposure to PDE5 inhibitors was defined as the date of first dispensation."

This is problematic because the authors are conditioning exposure based on an event that happens in the future i.e. a patient can only be exposed if they have 2 dispensations, but they start follow up at their first dispensation, and median time between first and second dispensation is 4 months. It is possible that this causes considerable time related bias. To get around this, authors should start follow up at either first dispensation and not require patients to have 2 dispensations, or if they insist they require patients to have 2 dispensations, then follow up should start at the second dispensation.

2) Following on from my original comment 10 about immortal time, the authors state that they used a matched cohort design to limit the impact of immortal time bias. Matching patients at baseline will not limit immortal time bias. The only way to do this is through the correct specification of time zero (an example of which is in the comment above in relation to the correct specification of start of follow up).

Reviewer #2 (Remarks to the Author):

The authors addressed the reviewer's comments appropriately. The reviewer has no additional comments.

Dear editor and reviewers,

Thank you for your comments on our paper "Phosphodiesterase-5 inhibitors use and risk for mortality and metastasis among male patients with colorectal cancer ". We have carefully read your comments and modified the text accordingly. All the main changes in the text are shown with red color. Please see our point-by-point responses to the comments below.

Reviewer #1:

The authors have clearly put a lot of effort into revising the manuscript following my original comments. The revisions have considerably changed some of the original methodologies. For example, confounders have now been defined at baseline, rather than their status before the end of follow up. However, I still have some considerable worries about their methodological choices.

1) Following my original comment 4, the authors state: "To overcome non-adherence of PDE5 inhibitors use, we identified patients with two or more dispensations of PDE5 inhibitors after diagnosis defined as users. Exposure to PDE5 inhibitors was defined as the date of first dispensation."

This is problematic because the authors are conditioning exposure based on an event that happens in the future i.e. a patient can only be exposed if they have 2 dispensations, but they start follow up at their first dispensation, and median time between first and second dispensation is 4 months. It is possible that this causes considerable time related bias. To get around this, authors should start follow up at either first dispensation and not require patients to have 2 dispensations, or if they insist they require patients to have 2 dispensations, then follow up should start at the second dispensation.

>>> We appreciate that the reviewer pointed out the potential impact of immortal time between the first and second dispensation (the median interval was 4 months) which was neglected in our analysis. We agree that follow-up should start at the date of second dispensation. After re-defining the start of follow up, CRC patients with PDE5 inhibitor

use were still at a significant lower risk of metastasis and mortality. Please see the revised manuscript.

2) Following on from my original comment 10 about immortal time, the authors state that they used a matched cohort design to limit the impact of immortal time bias. Matching patients at baseline will not limit immortal time bias. The only way to do this is through the correct specification of time zero (an example of which is in the comment above in relation to the correct specification of start of follow up).

>>> We also re-defined the start of follow up to the date of second dispensation in matched cohort design, we believe that the matched cohort study design could be an another efficient method to limit immortal time bias as suggested by some other scientists and used in many pharmacoepidemiological studies [1-4]. As shown in the Supplementary figure 2, the matched comparisons without using PDE5 inhibitors were randomly selected for each PDE5 inhibitor user conditional on the same year of birth and CRC diagnosis, and CRC patients who used PDE5 inhibitor and the matched comparisons were followed at the same time (date of second dispensation in users and the same date in the corresponding comparisons).

References

1. Hong JL, Meier CR, Sandler RS, et al. Risk of colorectal cancer after initiation of orlistat: matched cohort study. *BMJ* 2013;347:f5039.
2. Hung SC, Chang YK, Liu JS, et al. Metformin use and mortality in patients with advanced chronic kidney disease: national, retrospective, observational, cohort study. *Lancet Diabetes Endocrinol* 2015;3:605-14.
3. Cea Soriano L, Gaist D, Soriano-Gabarro M, et al. Low-dose aspirin and risk of intracranial bleeds: An observational study in UK general practice. *Neurology* 2017;89:2280-2287.

4.Ji J, Sundquist J, Sundquist K. Cholera Vaccine Use Is Associated With a Reduced Risk of Death in Patients With Colorectal Cancer: A Population-Based Study. *Gastroenterology* 2018;154:86-92.e1.

Yours sincerely,

Wuqing Huang & Jianguang Ji

On behalf of co-authors

REVIEWERS' COMMENTS:

Reviewer #1 (Remarks to the Author):

All my points have been addressed, thank you.